# A Cyanobacteria Enriched Layer of Shark Bay Stromatolites Reveals a New *Acaryochloris* Strain Living in Near Infrared Light

**DOI:** 10.3390/microorganisms10051035

**Published:** 2022-05-17

**Authors:** Michael S. Johnson, Brendan P. Burns, Andrei Herdean, Alexander Angeloski, Peter Ralph, Therese Morris, Gareth Kindler, Hon Lun Wong, Unnikrishnan Kuzhiumparambil, Lisa M. Sedger, Anthony W. D. Larkum

**Affiliations:** 1School of Medical Sciences, University of New South Wales, Sydney 2052, Australia; 2School of Life Sciences, University of Technology Sydney, Sydney 2007, Australia; lisa.sedger@wimr.org.au; 3School of Biotechnology and Biomolecular Sciences, University of New South Wales, Sydney 2052, Australia; brendan.burns@unsw.edu.au (B.P.B.); h.l.wong@unsw.edu.au (H.L.W.); 4Australian Centre for Astrobiology, University of New South Wales, Sydney 2052, Australia; therese.morris@blueearthscience.com.au (T.M.); garethseankindler@gmail.com (G.K.); 5Climate Change Cluster, University of Technology Sydney, Sydney 2007, Australia; andrei.herdean@uts.edu.au (A.H.); peter.ralph@uts.edu.au (P.R.); unnikrishnan.kuzhiumparambil@uts.edu.au (U.K.); a.larkum@sydney.edu.au (A.W.D.L.); 6School of Mathematical and Physical Sciences, University of Technology Sydney, Sydney 2007, Australia; alexander.angeloski@uts.edu.au; 7School of Earth and Planetary Sciences, Curtin University, Perth 6102, Australia; 8Department of Aquatic Microbial Ecology, Institute of Hydrobiology, Biology Centre of the Academy of Sciences of the Czech Republic, 37005 České Budějovice, Czech Republic; 9Centre for Virus Research, Westmead Institute for Medical Research, University of Sydney, Sydney 2145, Australia

**Keywords:** cyanobacteria, *Acaryochloris*, chlorophyll *d*, near infrared, stromatolite, shark bay

## Abstract

The genus *Acaryochloris* is unique among phototrophic organisms due to the dominance of chlorophyll *d* in its photosynthetic reaction centres and light-harvesting proteins. This allows *Acaryochloris* to capture light energy for photosynthesis over an extended spectrum of up to ~760 nm in the near infra-red (NIR) spectrum. *Acaryochloris* sp. has been reported in a variety of ecological niches, ranging from polar to tropical shallow aquatic sites. Here, we report a new *Acarychloris* strain isolated from an NIR-enriched stratified microbial layer 4–6 mm under the surface of stromatolite mats located in the Hamelin Pool of Shark Bay, Western Australia. Pigment analysis by spectrometry/fluorometry, flow cytometry and spectral confocal microscopy identifies unique patterns in pigment content that likely reflect niche adaption. For example, unlike the original *A. marina* species (type strain MBIC11017), this new strain, *Acarychloris LARK001*, shows little change in the chlorophyll *d/a ratio* in response to changes in light wavelength, displays a different Fv/Fm response and lacks detectable levels of phycocyanin. Indeed, 16S rRNA analysis supports the identity of the *A. marina* LARK001 strain as close to but distinct from from the *A. marina* HICR111A strain first isolated from Heron Island and previously found on the Great Barrier Reef under coral rubble on the reef flat. Taken together, *A. marina* LARK001 is a new cyanobacterial strain adapted to the stromatolite mats in Shark Bay.

## 1. Introduction

Chlorophyll *d* (Chl *d*) was discovered in 1943 and was first attributed to marine macrophytic red algae ([1] and references therein). It was the fourth chlorophyll to be discovered, hence the name attribution. It differed from chlorophyll *a* (Chl *a*) in having a formyl group at C3 in ring A instead of a vinyl group. This small change meant that the Qy peak was red shifted (in vitro) from 665 nm to 696 nm. For the next fifty years, the existence of Chl *d* was debated and often relegated to a breakdown product resulting from changes during extraction [1]. More than half a century later, the cyanobacterium *Acaryochloris marina* (strain *Acaryochloris marina* gen. et sp. Nov., MBIC11017 (cyanobacteria) was isolated [2] and later confirmed to be an oxygenic photosynthetic prokaryote containing Chl *d* as its major photopigment [3]. It was subsequently recognised that the Chl *d* found in red algal species originated from colonial *Acaryochloris* spp. living on the surface of the algae [4,5,6].

A second isolate was later obtained as a free-living epiphytic cyanobacterium present on the surface of a number of red algae [6]. Thus, *Acaryochloris* grows as a free-living organism, and a large part of the presence of this species in didemnid ascidians is due to an epiphytic growth [4,5]. A third isolate belonging to the *A. marina* clade (strain CCMEE5401) was characterized in 2005, isolated from a unique site at the Salton Sea saltwater lake in California, USA [7]. However, a comparison of the 16S rRNA sequences of both strains indicates firstly that they are highly similar (99.2% nucleotide sequence identity) and also that the *A. marina* clade contains a characteristic small-subunit 16S rRNA gene insertion likely originating from a b-proteobacterium; hence, *Acaryochloris* is a proteobacterial-cyanobacterial hybrid entity based on 16S rRNA analysis [7].

Several additional strains of *Acaryochloris marina* have been isolated and cultured: *A. marina* MBIC11017 isolated from the squeezed extract of didemnid ascidian in Palau Island [2,3], which is likely the same epiphytic organism also recovered elsewhere [4,5], *A. marina* AWAJI-1 isolated as an epiphyte from red algae in Japan [6], *A. marina* sp. CCMEE 5410 isolated from a eutrophic hypersaline lake in Salton Sea, California [7], *A. marina* sp HICR111A isolated from dead coral on the reef flat at Heron Island, Australia [8] and the isolated strains, *A**. marina* MPGRS1—an epiphyte from the red alga *Gelidium caulacantheum* collected from Georges River, Australia [9] and another from a shaded calcified substrate on a coral reef [10]. Of note, *A. marina* has also been found in a number of habitats [11,12] including Antarctica [13], although 16S rRNA data are lacking from these more recently isolated strains.

Unlike most other photooxygenic organisms, *A. marina* uses Chl *d*, instead of Chl *a*, as a main pigment in the reaction centres and for photosynthetic light-harvesting. Chl *d*, differs from Chl *a* at longer wavelengths by absorbing in the near-infrared (NIR) region (approx. 710 nm in vivo) rather than the red region of the visible spectrum. Chl *d* was at first believed to be unique to the genus *Acaryochloris* [3,5]. However, Chl *d* has been found in a range of cyanobacteria [14,15,16,17,18,19], including some cyanobacteria that possess both Chl *d* and Chl *f* [16,19]. Investigation of the gene cluster FaRLiP (Far Red Light Photoacclimation) indicates that this is a genetic locus responsible for the synthesis of Chl *d* and Chl *f* in several cyanobacteria—correlating with the capability of growing in far-red light [16] (Op. Cit). In cyanobacterial species containing FaRLiP, Chl *d* forms only a minor part of the total chlorophyll, yet it has been shown to be present in the reaction centre of photosystem II in *Chroococcidiopsis* [20]. Nevertheless, all analyses of the 16S rRNA sequence data of *A. marina* cyanobacteria from environmental DNA samples that have been examined to date indicate that these photosynthetic bacteria are very widespread in nature [21,22,23,24]. Despite Chl *d* having been discovered in 2003/2004 in a range of other cyanobacterial genera (op. cit.), it is only recently that a second species, *Acaryochloris thomasi* [22], has been identified. This species differs from all other isolates of *Acaryochloris* in that it completely lacks Chl *d* but rather has monovinyl-Chl *a* and *b* content similar to that in *Prochloron* and *Prochlorothrix*.

The genome of *A. marina* MBIC11017 (type strain) was recently analysed and found to be one of the largest cyanobacterial genomes, comprising 8.3 million nucleotides distributed between one master chromosome of 6,503,724 base pairs and 9 plasmids [23]. This unusual genome organization, particularly the fact that approximately 25% of its genes reside in plasmids, gave rise to the idea of an increased plasmid-based lateral gene transfer within this genus [23]. Furthermore, the genes located within the plasmid components include those that code for small-molecule biosynthesis, central or intermediary metabolism, energy metabolism, signal transduction, DNA metabolism, transcription, protein synthesis/fate and surface-associated features [23]. Presumably, the plasmids have the capacity for lateral gene transfer to confer an environmental adaptation advantage. Despite these intriguing features of the type strain, the *Acaryochloris* species as a whole is still poorly described genetically and phenotypically using currently available technologies.

Here, we have characterised a strain of *A. marina* isolated from the 4–6 mm layer of smooth stromatolites from Nilemah, Hamelin Pool of Shark Bay in Western Australia. This layer receives mainly NIR radiation [24] consistent with the need for specialized NIR absorbing chlorophylls, such as Chl *d* and/or Chl *f*. This study therefore characterises this unique *A. marina* strain from Australian stromatolites and compares it with the *A. marina* type strain (MBIC11017), using 16S phylogenetic typing and pigment analysis by spectrometry/fluorometry, flow cytometry and spectral confocal microscopy to reveal its unique pattern in pigment distribution that likely reflects stromatolite niche adaption.

## 2. Materials and Methods

### 2.1. Site Description, Sampling and Mat Sectioning

Sampling was undertaken at Nilemah, on the southern shore of the Hamelin Pool (26°27′336″ S, 114°05.762″ E), at 1200 h on 21 June 2015 (see Figure 1A). This is an area of high salinity due to the isolation of the Hamelin pool from the rest of Shark Bay as a result of a sill at the entrance to the Hamelin Pool (salinity 66–72 PSU). At the time of sampling, the water temperature was 20.1 °C, the salinity was 67.4 PSU (Practical Salinity Units), i.e., a hypersaline environment, and the pH was 8.13. The samples were collected under DBCA license FO25000229-2, and the research was conducted on Malgana country and in consultation with the traditional owners by AWDL. A sample was collected and transported to UTS for the various analyses. It was stored at 4 °C in the dark prior to analysis. The mat types analysed here were designated smooth mats, as they are extensively distributed in Shark Bay and have been examined in detail at the taxonomic and metagenomic levels [25,26,27,28]. A map showing the location of sampling is shown in Figure 1A. Sections of mat approx. 5 cm × 5 cm and 2 cm deep were obtained with a sharp knife and placed in waterproof plastic containers for air transport to the University of Technology, Sydney within 24 h; they were then sored at 4 °C in the dark until examination within 2 months. For confocal microscopy, thin sections (Figure 1B) of the mat material (approx. 2 mm thick) were obtained and placed on slides, which were examined with a Nikon A1 Confocal Microscope equipped with a spectral detection capability from 420–750 nm in 10 nm bandwidths. These mat samples are considered representative of stromatolitic mats in Shark Bay, as shown by previous studies. Cultures of *A. marina* MBIC11017 were obtained from the laboratory of Prof. Min Chen at the University of Sydney.

### 2.2. Isolation and Culturing

The overall isolation and culturing strategy is summarised in Figure 1B. Briefly, a thin slice (2 mm thick) representing the green layer (4–6 mm) from the upper surface was excised with a scalpel blade. This layer was transferred to a sterile 35 mm glass bottom petri dish (imaging grade WPI dish) and imaged by confocal microscopy. After imaging the section, it was transferred to artificial seawater supplemented with BG11 liquid broth and grown in NIR light to enrich far red absorbing microorganisms. The enriched culture was transferred to agar plates containing artificial seawater supplemented with BG11 so as to obtain single colonies under NIR conditions. A single colony was then sub- cultured into artificial seawater supplemented with BG11 liquid broth and further grown in NIR light to an appropriate cell mass for analysis. Purity was checked by 16S rRNA sequencing. Once purity reached greater than 95%, flow cytometry, confocal microscopy, spectroscopy and HPLC were employed to analyse the culture material.

### 2.3. Liquid Culture Enrichment

Culture aggregates from the 4–6 mm layer of several stromatolite samples were ground up with a 10 mL *Potter*-*Elvehjem* (*glass-teflon*) hand homogeniser with 5 strokes and were resuspended in BG11 medium (Sigma 73816) dissolved in seawater (at 30 PSU) and incubated in NIR radiation (720 nm, 10 µJ m^−2^ s^−1^) for up to 12 weeks. Further subculturing involved homogenising 10 mL of the existing culture and diluting the homogenate into new medium at a 1/10 ratio. Liquid cultures were then grown in NIR light (720 nm, 10 µmols m^−2^ s^−1^), with shaking at 60 rpm at room temperature (23 °C).

### 2.4. Culturing on Agar Plates

A 500 µl of aliquot liquid culture homogenate was spread onto agar plates (artificial seawater supplanted with BG11 and 15% agar) and incubated as above in NIR or white light. Single green colonies were selected and re-inoculated with a sterile loop onto BG11 supplemented seawater agar plates and grown in white light (WL) (100 µmols m^−2^ s^−1^) or NIR radiation (720 nm, 10 µmols m^−2^ s^−1^) for up to 12 weeks (Figure 1B).

### 2.5. Fluorescence Microscopy

Hyperspectral confocal microscopy was used to determine emission spectra matched to cell morphologies. Mat samples sections, small aliquots of liquid culture or colonies from agar plates were laid onto WPI Fluoro-dishes (coverslip thickness 0.017 mm), and images were scanned on a Nikon A1 Confocal microscope using the 458 nm laser with emission collected through the spectral detector (up to 750 nm) in order to obtain a lambda stack. Cells of interest were highlighted by the Regions of Interest (ROIs) function, and the emission spectra were collected for those cells.

For the 3D imaging of isolates from liquid culture, cells were fixed in 4% Paraformaldehyde/PBS and mounted on a 0.017 mm coverslip in a prolong glass (NA1.520) mounting medium (ThermoFisher, Waltham, MA, USA) and imaged on a Leica Stellaris 8 Confocal microscope with a 63× oil object (NA1.4). The excitation and emission settings were set to detect phycocyanin fluorescence (excitation 620 nm, emission 675/10 nm) and Chl *d* fluorescence (excitation 708 nm, emission 720/10 nm). A step size of 100 nm was used to collect the whole volume of the cells. For morphometric analysis, the fixed and mounted cells (above) were imaged on an Olympus BX51 epifluorescence microscope fitted with a 100× oil immersion objective (NA1.3) and an Olympus DP73 camera. Both brightfield and fluorescence images (excitation 500 nm–530 nm emission 575 nm LP) were collected. Cell shape (aspect ratio) and size (diameter) were determined by using Image J—FIJI [29]. For each isolate, a threshold level was set for the red auto-fluorescence, where the fluorescence perimeter of the cells matched the brightfield perimeter. The segmented cells were made into a binary image, and any joining cells were separated by water-shedding. The “analyze particles …” function was then used to determine the shape (aspect ratio). Further segmentation based on circularity (where non-round/elongated cells, i.e., cells with circularity less than 0.8, were discarded) was applied, and the diameter of these “round” cells was also determined using the “analyze particles …” function.

### 2.6. Absorbance and Fluorescence Spectroscopy

Slides of cultured *Acaryochloris sp.* were placed on glass microscope slides with coverslips and placed on an upright microscope stage (Nikon Ti, Tokyo, Japan) with a 40× (NA0.6) objective at room temperature. The cells were brought into the focal plane under white light. The absorbance of the light that passed through the cells was measured with a spectrometer (Avantes Avaspec with a spectral resolution of 0.4 nm) connected via an optical cable to one of the oculars of the microscope. An area devoid of cells was used to collect the background spectra that was later subtracted in the image analysis processing.

### 2.7. Excitation Emission Matrices

Liquid cultures were placed into the wells of a 96 welled (black walled) plate, and the fluorescence emissions at every wavelength from 620–760 nm were measured at every excitation wavelength from 300–750 nm in a fluorescence plate reader (Tecan Infinite M1000 Pro, Tecan, Männedorf, Switzerland). The raw data were exported to Microsoft Excel, and heatmaps were created as described by [30].

### 2.8. HPLC Pigment Analysis

High performance liquid chromatography (HPLC) was used to determine the concentrations of Chl *a* and *d* in the *Acaryochloris* cultures grown in either white light or NIR cultures. Each culture was pelleted at 5000 g for 10 min to collect the biomass for pigment signal detection. The extraction of samples were carried out following [31], with slight modifications. Chilled acetone was added to the pelleted biomass, probe sonicated (30 s at 50 W on ice), vortexed three times for 30 s each under cold, dark conditions to limit pigment degradation and then stored at 4 °C overnight. The pigment extracts were then filtered through a 0.2 µM PTFE 13 mm syringe filter and stored at −80 °C until analysis. An Agilent 1290 HPLC system equipped with a binary pump with an integrated vacuum degasser, thermostat-controlled column compartment modules, an Infinity 1290 autosampler and a PDA detector was used for the analysis. Column separation was performed using a Zorbax Eclipse C18 HPLC 4.6 mm × 150 mm column (Agilent, Santa Clara, CA, USA) using a gradient of ammonium acetate (0.01 M), methanol, acetonitrile and ethyl acetate. A sandwich injection approach was set using the auto injector program, where the mixture of MeOH: ACN (8:2) and the samples were drawn alternatively in the sequence, 100:60:100 (µL), and then mixed in the loop and injected. A complete pigment spectrum from 270 to 780 nm was recorded using a PDA detector with 3.4 nm bandwidth. The following wavelengths were used to monitor the chromatogram: 406, 440, 660, 696 and 706 nm.

### 2.9. 16SSU rRNA Analysis

DNA was extracted using the DNeasy PowerBiofilm Kit DNA extraction kit (QIAGEN, Hilden, Germany) according to the manufacturer’s instructions. Biological duplicates were extracted separately and then pooled prior to sequencing to minimise potential heterogeneity biases. Purified DNA samples were sequenced on an Illumina MiSeq for 16S V1–V3 amplicon (27F-519R) on a v3 2 × 300 bp run by the Ramaciotti Centre for Genomics (UNSW Sydney, Australia). Sequences were processed through the Mothur pipeline [32] with default parameters, with searches of representative sequences against the SILVA [33] database. For the most abundant sequence in the cultures, representative sequences were then aligned against Cyanobacteria 16S rRNA sequences retrieved from the SILVA and NCBI databases [34], with Chloroflexi as the outgroup. A maximum likelihood tree was constructed with IQ-TREE2 with 1000 bootstraps, and the best-fit model TIM3+F+I+G4 was chosen by ModelFinder. The tree was visualized using iTOL [35].

### 2.10. Flow Cytometry

*Sample acquisition and instrument settings*. Aliquots of *Acaryochloris*—grown cultures in BG11/seawater (grown to a purity of 95% by rRNA sequencing as described above in Section 2.2)—were prepared for flow cytometry analysis by executing five strokes of a *Potter*-*Elvehjem* (*glass-teflon*) hand homogenizer tube to break up cyanobacterial aggregates. Analysis was carried out on an LSRII flow cytometer (Becton Dickenson, BD Biosciences, Franklin Lakes, NJ, USA) with DIVA acquisition software (version 8.0.2). First, FSC-A/SSC-A two-parameter dotplots were viewed with logarithmic scale settings. Sterile normal saline was acquired for 1 min pre-cyanobacterial sample acquisition. This permitted the detection of electronic noise events and permitted subsequent gating on larger events of cyanobacteria (R1) from the liquid cultures. Next, cell events were collected for each culture sample before again running sterile saline, also for 1 min. Voltages were set such that electronic noise and any non-fluorescent cyanobacteria were located in the bottom left-hand panel of the two-parameter dot-plots of all fluorescence detection channels. Thus, natural fluorescence was evident as events with higher laser-excited fluorescent emission and without any staining. The data were saved and exported as FCS3 data files.

*Cytometry Data Analysis.* The samples’ file events were set to the same minimum 80,000 number using the DownsizeSampleV3 plugin. The samples were first viewed as two-parameter FSC-A/SSC-A dot-plots. Uncompensated data were also concatenated for subsequent analysis for SSC-A distribution (or FSC-A distribution) versus sample ID to confirm cyanobacterial events (R1) as distinct from electronic noise events (*). Events in R1 were assessed as FSC-A data via histogram overlays—gating on very small cells and debris, small cells and large cells. Uncompensated two-parameter fluorescence data excited by the 640 nm red laser were chosen for the fluorescent analysis, as this represented the presence or abundance of phycocyanin (detected via the 660/20 nm bandpass filter in detector array position C) and Chl *d* (detected via the 730/45 nm bandpass filter in detector array position B). All cytometry analysis and data overlays were performed with FlowJo (version 10.8.1, Flowjo LLC, Ashland, OR, USA), and multi-component data figures were prepared in Microsoft Powerpoint for Mac (version 16.57, Microsoft, Redmond, WA, USA).

### 2.11. Fluorometry by Pulsed Amplitude Modulated (PAM) Fluorescence

The quantum yield of photosystem II (F_v_/F_m_) was determined by the MAXI PAM model IMAG-MAX/L (Walz, Effeltrich, Germany) fluorometer. The instrument was fitted with blue actinic light emitting diodes (LEDs) with a peak wavelength of 450 nm which were used for applying the saturation pulse for the determination of F_m_ and for the determination of F_o_. The samples were dark adapted for 10 min prior to the measurement. After image acquisition, data processing was performed with the ImagingWin v2.56p instrument software (Walz, Effeltrich, Germany) [36].

### 2.12. Cell Counts

The growth rate was measured from liquid cultures using a Haemocytometer and an Olympus BX51 microscope fitted with a 20× (0.6 NA) objective. Prior to counting, the cell clumps were broken up by a potter Elvehjem-Potter homogeniser. The data were analysed in Graph Pad PRISM using the exponential growth curve function. 

## 3. Results

### 3.1. Location

Cyanobacteria were isolated from stromatolitic mats located in the hypersaline environment at Nilemah in the Hamelin Pool (see Section 2.1 and Figure 1A).

### 3.2. Confocal Imaging of Mat Material

Hyperspectral confocal images were obtained of the mat at 4–6 mm from the upper surface. Both coccoid and filamentous cyanobacterial cells were visualised. Emission wavelengths evident from the filamentous cells exhibited a peak at 745–750 nm, indicating Chl *f* (Figure 2A), and coccoid cells with a peak at approximately 735 nm, indicating Chl *d* (Figure 2B). The filamentous cells with the 745–750 nm peak emission also had a significant peak in the red at 680–685 nm, indicating the presence of significant amounts of Chl *a* (Figure 2A). There were also coccoid cells with a spectral emission profile indicative of mainly Chl *a* (Figure 2A—data not shown). Hyperspectral confocal imaging of an initial culture enrichment in NIR indicated the presence of coccoid cells with an emission peak of 735 nm, filamentous cells with an emission peak of 728 nm and diatoms with an emission peak of 710 nm (Figure 2C).

Present in the confocal microscopy analysis of the mat material were several other bacteria, some with near far-red shifted peaks in their chlorophylls (data not shown). The most abundant of these was what we identified as *Chroococcidiopsis sp*., which possesses Chl *f* with a fluorescence emission peak at ~750 nm at RT. This *Chroococcidiopsis sp*. has not been previously reported to have been isolated to purity in in vitro culture. Similarly, another cyanobacterium with an apparent red-shifted chlorophyll is *Spirulina sp*., which also has not been isolated to purity to date. Moreover, the 16S genotype sequence data confirm the presence of both of these microorganisms in stromatolite material [23]. Furthermore, the 16S data presented here indicate the presence of other phototrophic bacteria, *Phormidia*, *Halomicronema* and *Cyanobium*, all of which have been detected microscopically (data not shown). Microalgae (chloroplast) are also evident in the 16S data, and these were evident microscopically in our initial enriched cultures (Figure 2C). The presence of Caenarcaniphilales (Malainabacteria) is also of interest. The Melainabacteria are a newly described, non-photosynthetic sister phylum to the Cyanobacteria. They are often found in aphotic environments such as human and animal guts, grassland soil and wastewater treatments [37], and it is interesting that they have been detected in the attenuated light conditions associated with the 4–6 mm deep layer studied here.

### 3.3. Obtaining a Purified Isolate and Culture Morphology

Sub-sampling of the enriched culture resulted in the growth of the coccoid forms as colonies on agar plates in NIR. Further subculturing of agar plates in NIR of single colonies led to the isolation of single colonies on agar plates (Appendix A). The MBIC11017 type strain had a propensity to form large clusters, compared with the LARK001 strain, when grown in liquid culture in NIR (Appendix A). The LARK001 strain exhibited a yellow green colour when grown on agar or when grown in liquid culture, whereas the MBIC11017 strain exhibited a darker blue-green colour in the same conditions.

### 3.4. HPLC Analysis 

HPLC analysis revealed that both the LARK001 strain and MBIC11017 strain possess Chl *d* as their major photosynthetic pigment. The Chl *d* content in the LARK001 strain remained consistent in response to light conditions, with Chl *d* making up 93.4% of the chlorophyll content in white light and 92.9% in NIR (Appendix A). However, the MBIC11017 strain showed a marked difference in chlorophyll content in response to light conditions, with 91% Chl *d* content evident in cells grown in white light, which increased to 96.4% Chl *d* in cells grown in NIR (Appendix A).

### 3.5. Sequence Identity

The 16S DNA sequence identity of cultured *A. marina* LARK001 cyanobacteria from Shark Bay stromatolites indicated that it forms a clade closely linked to *A. marina* HICR111A, first isolated from Heron Island, as demonstrated by maximum likelihood phylogenetic tree analyses of representative species (Figure 3A); 98.67% similarity. Indeed, whilst the LARK001 isolate is clearly an *A. marina* cyanobacteria, it clusters away from the *A. marina* MBIC11017 type strain (Figure 3A). Moreover, although not entirely genetically pure, this LARK001 in vitro cultured isolate is over 95% *A. marina* and can sustain in vitro growth when isolated together in co-culture with up to three other species (Figure 3B). In the initial enrichment, 16S rRNA analysis indicated the presence *Phormidesmiales*, *Oxyphotobacteria* and *Halomicronema*, along with *Acaryochloris*. In the late enriched culture, a 16S rRNA gene analysis revealed the presence of *Halomicronema* and Chloroplasts (most likely diatoms), which were both also found by confocal microscopy in an enriched culture—see Figure 2C. It is interesting to see the presence Caenarcaniphilales in the late enriched culture. These are a newly described order of organisms, belonging to the phylum Melainabacteria, that share phylogenetic and structural similarities with cyanobacteria but lack the ability to photosynthesise. Nevertheless, this new Australian *Acaryochloris* strain constituted 95% of the late enriched culture and is distinctly phylogenetically distant from the clade containing the MBIC11017 type strain.

### 3.6. Spectroscopic Analysis

Next an absorbance spectral analysis was performed on cultures of the *A. marina* MBIC11017 strain and the *A. marina* LARK001 strain. The absorbance peak of *A. marina* MBIC11017 was 710 nm, whereas that of *A. marina* LARK0001 was at 704 nm (Figure 4B). However, whilst *A. marina* MBIC11017 additionally has a distinct absorbance peak at 610–640 nm (attributed to phycocyanin), this was not detectable in the new *A. marina* LARK001 (Figure 4A).

The fluorescence emission spectra of the two strains differed markedly (Figure 4C): the *A marina* MBIC11017 had a major emission peak at ~724 nm, whereas the *A. marina* LARK001 had a major peak at 735 nm (Figure 4C). Excitation emission scanning further highlighted the different profiles of these strains (Figure 4D). Notably, an additional fluorescence emission peak at 650 nm (attributed to phycocyanin) is present in the *A. marina* MBIC11017 strain, and there is no evidence of this in the LARK001 (Figure 4C). Finally, spectral confocal imaging analysis also confirmed that Chl *d* fluorescence emission peaks of 715 nm were evident for the *A. marina* MBIC11017, but in *A. marina* LARK001 (Figure 4; panels D,F), an additional fluorescence emission peak of 735 nm appears to predominate. Despite differences in the spectral characteristic at a cellular in vivo level, there were no such differences observed in the absorbance profiles in the HPLC Chl *d* fraction from each strain (Panel F).

### 3.7. Flow Cytometry and Morphometric Analysis

In vitro cultures of *A. marina* MBIC11017 and LARK001 were examined by flow cytometry after homogenisation to enable single cell cytometry. The small size and diverse culture shapes of these cyanobacteria are evident in forward and side scatter dot-plots, and the preparation procedure appears to produce very small cells and debris, but these are easily made distinct from electronic noise by comparing the files generated by saline (without cyanobacteria) and gating on side-scatter or forward scatter profiles (Figure 5A). Thus, the in vitro cultured cyanobacteria cells were detected in three general size groupings: very small, small and large (Figure 5A). Simple two-parameter uncompensated fluorescent data overlays of 640 nm red-laser excited emission profiles can easily profile both phycocyanin (evident on the 660/20 nm bandpass/detector channel) and Chl *d* (detected at the 730/45 nm detector channel) where distinct populations are evident (Figure 5B). A clear feature of this data is that the largest cells contain the most abundant Chl *d*, and the presence of phycocyanin in the MBIC11017 type strain easily separates the two strains (Figure 5C). In comparison, the small cells both contain low fluorescence in 660/20 nm and 730/45 nm, and the very small cells and debris are either weakly fluorescent or non-fluorescent in these channels (Figure 5C). The flow cytometry profiles also closely mirrored the images of two strains obtained by confocal fluorescent microscopy (Figure 5D), using the 620 nm laser excitation and collecting emissions at 675/10 nm for phycocyanin fluorescence and using laser excitation at 708 nm with emissions collected at 720/10 nm for Chl *d* (Figure 5D). Here, 3D rendering indicated that, in the MBIC11017 strain, the phycocyanin was present near the periphery of the cells, and Chl *d* was situated throughout the entire cell volume (Figure 5D), whereas no phycocyanin was present in the LARK001 strain.

Finally, both the cytometry side scatter profiles and confocal imaging strongly suggested that the MBIC11017 strain and the LARK001 strain may exhibit morphometry differences. Therefore, the fluorescence microscopy image data were analysed with respect to aspect ratio and diameter, and here, the literal physical shape differences were confirmed in that the LARK0001 strain was statistically more spherical and larger in diameter than the MBIC11017 strain (Figure 5E). However, it should be taken into account that the MBIC11017 strain divides at a faster rate than the LARK001 strain, potentially resulting in a greater representation of elongated (non-spherical) forms.

### 3.8. Fluorometry

Pulse amplitude modulated (PAM) fluorescence measurements facilitated the determination of the ratio of variable fluorescence (F_v_) to maximum fluorescence (F_m_) and hence the potential quantum efficiency of photosystem II for each strain under each light condition. The F_v_/F_m_ values of the MBIC110117 strain did not appear to change significantly in response to the light condition. However, the LARK001 strain showed a significant drop in potential quantum efficiency in photosystem II when incubated in NIR as compared to the cells grown in white light (Figure 6 Panels A,B). The LARK0001 cells grown in NIR showed the lowest quantum efficiency (F_v_/F_m_) of the all strains/conditions tested, and this was shown to be significantly different from all other strain and condition combinations tested. Additionally, the LARK001 strain grown in NIR resulted in the slowest growth conditions, with a 53 day dividing time (Figure 6 Panel C). This is more than double the doubling time of any other combination of strain and light condition.

## 4. Discussion

We have isolated a new strain of *A. marina* from stromatolitic mats from the southern end of the Hamelin Pool, Shark Bay in Western Australia. This site is the one designated as available for research in the stromatolite area of the World Heritage Area. It is, in fact, the same site where all approved stromatolite research has been conducted over the last >20 years. All of the published and unpublished research, to date, indicates that, although there is some difference in calcium deposition, overall, these structures represent a typical photosynthetic community, as found on the columnar stromatolites 20 km East, further along the southern side of the Hamelin pool [38]. The new Australian stromatolite-derived *A. marina* strain described here (*A. marina* LARK001) differs in several ways from the MBIC11017 original strain. Firstly, in 16S rRNA phylogenetic analysis, the new strain is closest to the previously isolated strain *A. marina* HICR111A [8], from a shallow site on the reef flat at Heron Island, Southern Great Barrier Reef. Consistent with this genetic closeness, the *A. marina* LARK001 has a room-temperature (RT) fluorescence peak at 735 nm, indicating the abundance of Chl *d*. Indeed, both the HICR111A and LARK001 strains appear to possess Chl *d* that is conjugated with a specialized pigment protein that shifts the fluorescence emission to 735 nm and above. It has been shown that Chl *d* is bound by a light-harvesting pigment protein and is also present in the reaction centres of photosystem I and Photosystem II [20]. Instead, the MBIC11017 type strain has a shorter RT fluorescence emission of ~720 nm, indicating that *Chl d* is the major chlorophyll in both photosystem I and photosystem II and that its light-harvesting chlorophyll protein complex has a fluorescence emission at ~720 nm. Thus, it is likely that *A. marina* HICR111A and LARK001 contain different light-harvesting Chl *d*-binding proteins that cause a shifted emission at 735 nm and above at RT. Moreover, the light harvesting Chl pigment harvests NIR light efficiently and passes it on to photosystem II—and possibly to photosystem I as well, as evident in the PAM analysis. This interpretation is consistent with the HPLC analysis that confirmed only Chl *d* as the major pigment present within LARK001 cultures, with only a minor amount of Chl *a* (6–7%) being detected in this analysis (Appendix A). Future studies will investigate the effect of environmental light conditions on Chl *d* content relative to Chl *a*. Another similarity between the HICR111A and LARK001 strains of cyanobacteria is that they both appear to contain very little phycocyanin. Interestingly, LARK001 divides slowly in NIR compared to in white light (Figure 6C). On face value, we conclude that the two strains of *A. marina* LARK001 and HICR111A have different light harvesting properties compared to the MBIC11017 type strain. This is in spite of HICR111A occurring on a coral reef in a shaded rubble zone on the reef flat in Eastern Australia and LARK001 existing several millimeters below the surface of a stromatolitic mat in the Hamelin Pool, Western Australia—some 3800 km distant. Both sites are subtropical with respect to latitude (Heron Island 22° S and Hamelin Pool, Shark Bay 26° S) but are differentiated in terms of the ecosystems involved. However, the *A. marina* MBIC11017 type strain was isolated from a tropical site (Palau 7.5 °N) in the western pacific, which has a very different ecological niche compared to the location in which we isolated the *A. marina* LARK001. Interestingly, the Hamelin Pool in Western Australia, Heron Island and Palau are equidistant from each other. However, the *A. marina* MBIC11017 type strain grows much better in NIR light compared to LARK001. In natural light, the MBIC11017 strain is augmented by some orange-red light (600–700 nm) absorbed by phycocyanin. In this regard, the MBIC11017 strain grows on the underside of didemnid ascidians, where it receives mainly radiation that has been highly filtered by the *Prochloron didemni* that lives in the atrial cavity of the ascidians. Thus, it may generally receive NIR light in the 720 nm range, with incidental flecks of light in the orange red spectral region. In contrast, both the *A. marina* LARK001 and HICR111A strains seem to rely on NIR at 704–706 nm but also have the ability to exist in white light, where Chl *d* can absorb light in the blue region augmented by absorption by the carotenoids (460–540 nm) without any reliance on phycocyanin in the orange-red region of the visible spectrum. It remains possible that all *Acaryochloris* strains might also rely on the absorption of light by Chl *a*, which is present in all strains, even if in smaller quantities (Appendix A).

A subtle but important feature of our data is that both the flow cytometry and fluorescence microscopy (with morphometric analysis) demonstrated that the smaller cells within the in vitro *culture* of the *A. marina* LARK001 strain are slightly larger (1.48 µm–2.43 µm diameter) than the smaller cells in the MBIC111017 type strain cultures (1.41–2.32 µm diameter). This is in contrast to reports of the size of the HICR111A Heron Island strain, which, although also lacking in phycocyanin, is a much smaller at 0.75–1.00 µm [8]. Hence, the 16S genotyping similarities of the *A. marina* LARK001 strain to HICR111A also coincide with similarities in its morphometric features—despite the obvious and significant ecological niche differences described above. A deeper, full genome analysis of *A. marina* LARK001 is underway in our laboratory, which should better explain and define the similarities and differences between these three strains. This will provide a deeper understanding of the genetic limitations and molecular signatures of the adaptation capability—particularly the potential for serious sensitivities to habitat changes within the stromatolite microbial communities, which would be expected to occur with the ongoing climate change impacts, or insults from viral and other microbial predators.

## 5. Conclusions

We provide here the first physical and initial genetic characterisation of a unique strain of *A. marina* cyanobacteria, termed the LARK001, isolated from living Australian stromatolitic mats in Shark Bay, Western Australia. Using HPLC, spectroscopy, flow cytometry and microscopic techniques, we demonstrate that *A. marina* LARK001 contains plentiful amounts of Chl *d* and minor signatures of Chl *a* and lacks the phycocyanins that are present in the *A. marina* type strain MBIC11017. Despite the difference in ecological niche and the large location distance, *A. marina* LARK001’s closest genetic relative is the *A. marina* HICR111A strain from Heron Island, Eastern Australia. Taken together, these data suggest that the *A. marina* LARK001 strain is a novel cyanobacterial strain and the first to be isolated from a section of stromatolite where incident light is attenuated and enriched in NIR.

## Figures and Tables

**Figure 1 microorganisms-10-01035-f001:**
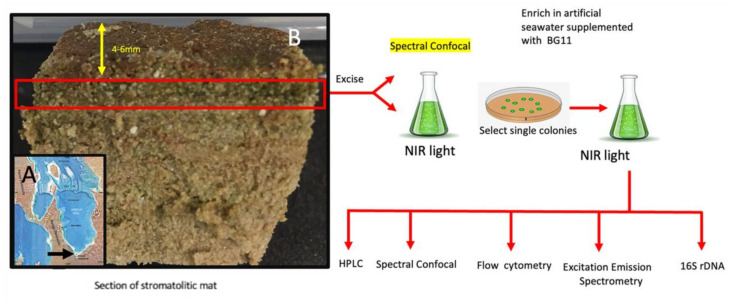
Sampling and enrichment: (**A**) Map showing the Hamelin Pool, with an arrow indicating the position of the sampling site at Nilemah. (**B**) Overview of sampling, culture enrichment and analysis methods.

**Figure 2 microorganisms-10-01035-f002:**
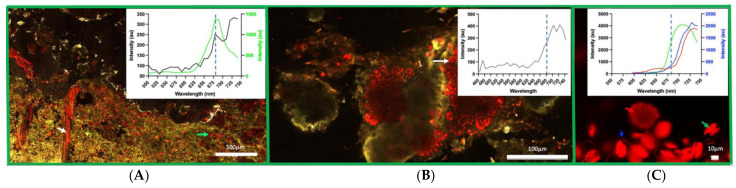
Confocal Imaging of the green layer. A side view of an excised segment randomly sampled from a smooth mat at Nilemah. For reference, the blue dotted line at 680 nm shows a typical Chl *a* emission. (**A**) A large section (1 cm × 0.6 cm) of the green layer imaged by hyperspectral confocal microscopy using a 458 nm laser with 10 nm continuous band pass emissions collected up to 750 nm. The red organisms consist of red and far red/NIR (700–750 nm) fluorescing pigments. Both filamentous and coccoid forms are observed. The filamentous forms (white arrow) contain a far-red emitting (750 nm peak) pigment that is most likely Chl *f*. The green arrow shows coccoid cells exhibiting a Chl *a* emission profile. Scale bar 500 µm. (**B**) Emission spectra of typical clusters of coccoid cells (white arrow) when excited by the 458 nm laser. Two peaks are evident at 704 nm and 724 nm, suggestive of the presence of Chl *d* (724 nm). Scale bar 100 µm. (**C**) The arrow shows the various microbial forms that were found in the initial NIR enrichment culture. The green arrow shows a diatom (emission peak 710 nm), the blue arrow shows a filamentous form (emission peak 730 nm) and the red arrow shows a cluster of coccoid cells (emission peak 730 nm). Scale bar 10 µm.

**Figure 3 microorganisms-10-01035-f003:**
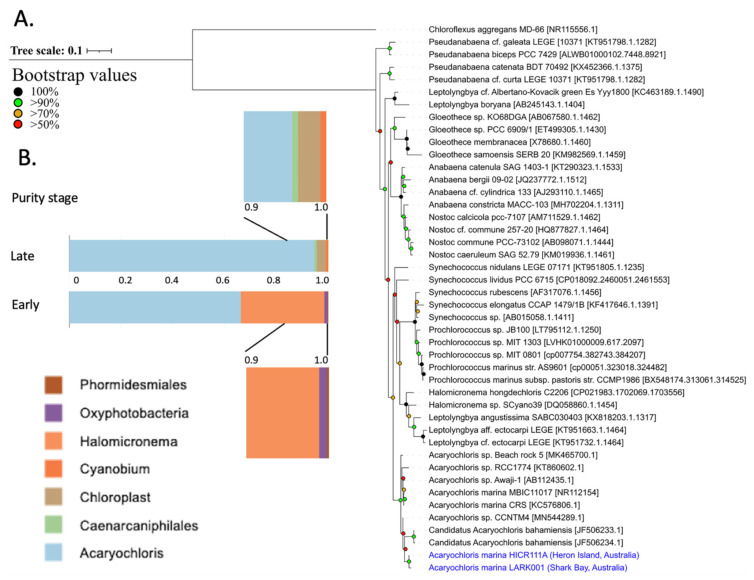
16S rRNA sequencing. (**A**) 16S rRNA gene phylogeny of the LARK001 isolate from showing the relationship to other isolates. (**B**) 16S rRNA gene sequence representation of the LARK001 isolate at different stages of culture purification.

**Figure 4 microorganisms-10-01035-f004:**
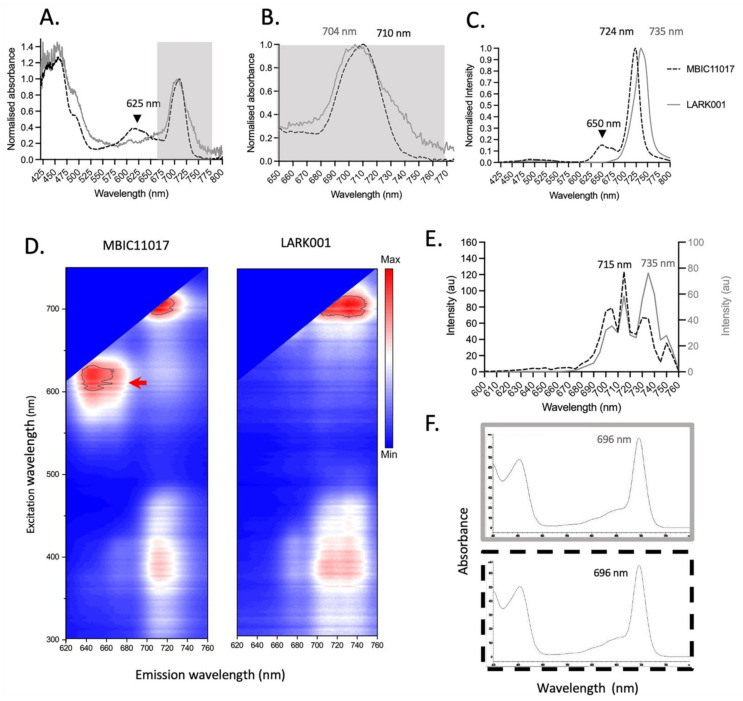
Absorbance spectra and fluorescence spectra of *A. marina* MBIC11017 vs. *A. marina* LARK001 grown in NIR. (**A**) Absorption spectra of both strains indicate differences between the two stains. At 625 nm (black arrow), a peak is present in the MBIC11017 strain that is not present in the LARK001 strain. Differences in the Chl *d* absorption region are also present (grey boxed region). (**B**) The grey boxed region from Panel A is magnified to further highlight the difference in peak Chl *d* absorption—704 nm for LARK001 and 710 nm for MBIC11017. (**C**) The emission profiles for each strain reveal a peak at 650 nm (black arrow) that is unique in the MBIC11017 strain, whilst the Chl *d* emission peaks differ substantially, with the peaks at 724 nm and 735 nm for MBIC11017 and at 735 nm for LARK001 strain. (**D**) An excitation emission matrix shows a unique spectral component (excitation max. approx. 625 nm and emission max. approx. 640 nm) in the MBIC11017 strain. The LARK001 strain exhibits two distinct emission profiles for Chl *d* at 718 nm (approx.) and 735 nm (approx.). Only one emission profile at approx. 718 nm is evident for the MBIC11017 strain. (**E**) Hyperspectral confocal scan (top panel) confirms the evidence of two emission peaks in LARK001, with the 735 nm peak being the major contributor in this strain. The same two peaks are evident in the MBIC11017 strain, but the 715 nm peak is the major peak. (**F**) The grey and black dashed panels reveal no difference in the absorbance profiles in the Chl *d* fraction that was purified by HPLC for each strain.

**Figure 5 microorganisms-10-01035-f005:**
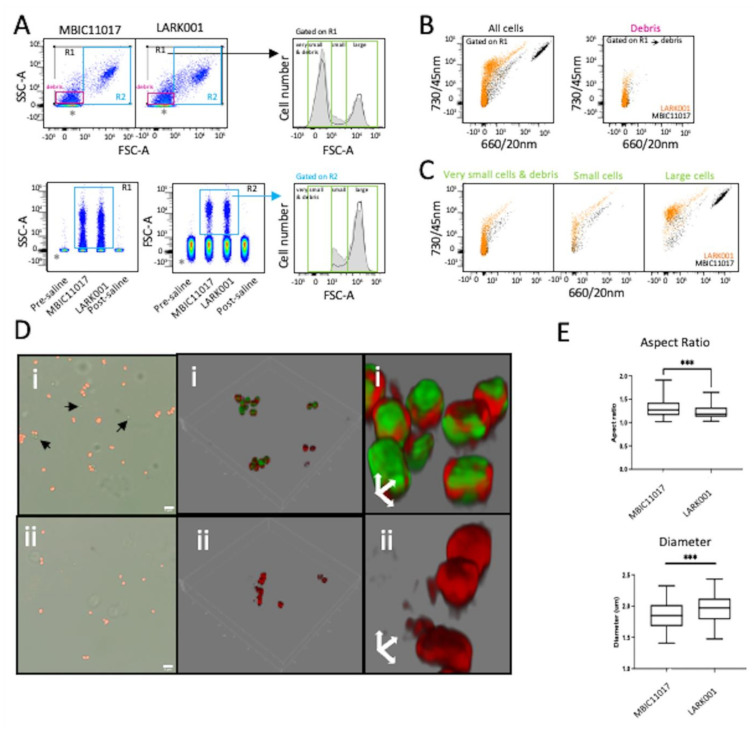
Cytometry and confocal microscopy analysis of A. *marina* type strains MBIC11017 and LARK001. (**A**) MBIC11017 and LARK001 cyanobacterial cells in FSC-A/SSC-A dot-plots indicating distinct cell populations. Also shown are concatenated SSC-A distribution data, confirming electronic noise events in pre- and post-saline samples (acquired for 1 min). Cell events in R1 exclude electronic noise (*) and demonstrate evidence of two sub-populations based on cell size: designated smaller (R2) and larger (R3) cyanobacteria cells in histogram overlays: MBIC11017 (black unfilled histogram) and LARK001 (grey filled histogram). Tiny debris events are also evident within the R1 gate, but size gating (R4) removes these events. The arrow indicates the shoulder of slightly larger cells within the R2 gated population. (**B**) Two-parameter dot-plots of natural fluorescence at 730/45 nm versus 660/20 nm in *A. marina* MBIC11017 (black) and LARK001 (orange) strains upon 640 nm red laser excitation. (**C**) Two-parameter dot-plots of fluorescence at 730/45 nm versus FSC-A for R2 and R3 gated events of *A. marina* MBIC11017 (black) and LARK001 (orange) strains, also upon 640 nm red laser excitation. (**D**) Widefield and confocal images of *A. marina* MBIC11017 (**i**) and *A. marina* LARK001 (**ii**) cells (scale bar is 5 mm); also shown are 3-D rendered and zoomed images of the same two strains. (**E**) Aspect-ratio (*n* = 107 for MBIC11017 and *n* = 94 for LARK001) and diameter analysis (*n* = 65 for MBIC11017 and *n* = 73 for LARK001) of cells from microscopy data acquired in D. *** *p* < 0.005 by a student *t*-test.

**Figure 6 microorganisms-10-01035-f006:**
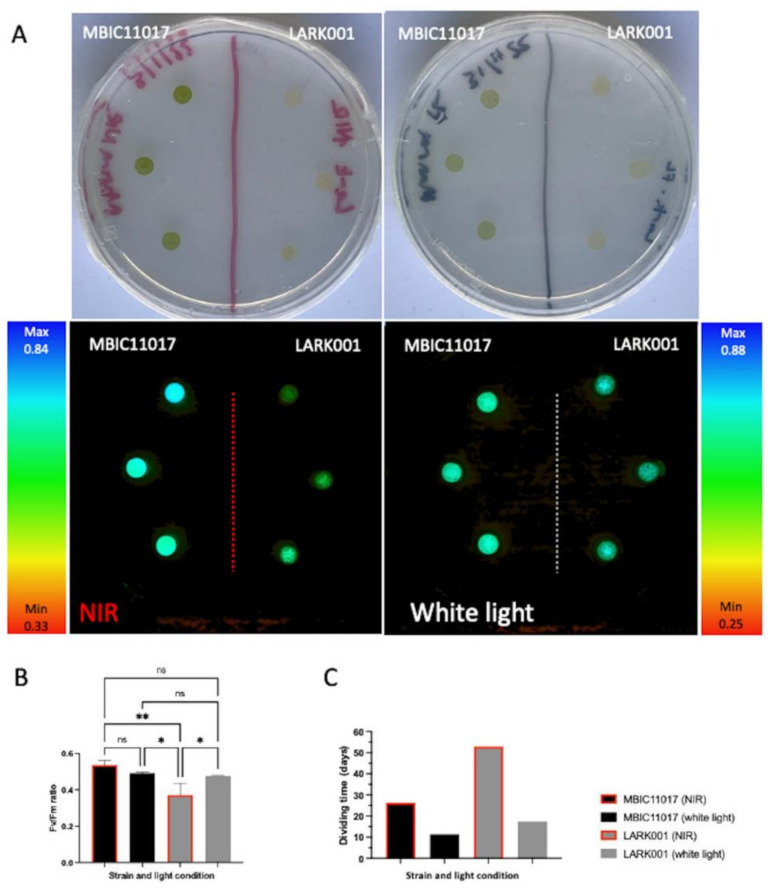
Fluorimetry measurement by PAM. (**A**) Clockwise from top left: Cultures grown for 3 weeks in NIR after 50,000/25 µL cells of each strain were added to the surface of the agar plate in triplicate. Cultures grown for 3 weeks in white light after 50,000/25 µL cells of each strain were added to the surface of the agar plate in triplicate. PAM imaging of the white light incubated cultures. PAM imaging of the NIR incubated cultures. (**B**) Fv/Fm measurements for the cultures and graph showing pairwise comparison * *p* < 0.05, ** *p* < 0.01. ns: not significant. (**C**) Doubling time of each strain in either white light or NIR conditions.

## Data Availability

All primary data are stored as a digital resource at UTS and are available upon request to the corresponding author. The DNA sequence has been uploaded as FASTQ sequences to MG-RAST.

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
