# Peer review of "A Cyanobacteria Enriched Layer of Shark Bay Stromatolites Reveals a New Acaryochloris Strain Living in Near Infrared Light"

_microorganisms, 2022, doi:10.3390/microorganisms10051035_

Round 1
Reviewer 1 Report
The manuscript by Johnson et al. describes a new Acaryochloris strain which was isolated from mats in Shark Bay, Australia. The authors use a combination of confocal microscopy, spectroscopy, 16S phylogenetics, HPLC and flow cytometry to describe the initial sample and their enriched isolate of Acaryochloris LARK001. Acaryochloris is an interesting genus of cyanobacteria due to its unique pigment composition with mainly chlorophyll d. As the authors point out, only a few strains have been isolated so far and the diversity of pigment composition is therefore largely unknown in this group. A recent study discovered an Acaryochloris strain that not even has chlorophyll d but instead chlorophyll b in addition to chlorophyll a. The new strain described by Johnsons et al. supports the diversity of pigment composition in this genus. Their finding that strain LARK001 might lack phycocyanin is intriguing and deserves further investigations in a separate study. The small change in chl d/a ratios upon changes in light (white light vs far-red light) is interesting but needs further support. Since the culture is not unialgal there might be significant changes in the cyanobacterial composition. 16S rRNA sequencing of samples grown under different light conditions could help to reveal the difference. Furthermore, the chl d/a ratio will most likely change according to the growth stage. This is currently not addressed in the manuscript.
General comment (maybe to editorial team): The article seems to be not correctly formatted. The keywords (line 31f), results (line 252ff) and author contributions (line 520ff) contain still the guidelines for the authors. Furthermore, the text on page 15 (line 616-339) is shifted to the right and you find several places where it is written “(Op Cit)” (e.g. 61, 76, 81). Since the figure and table in the supplementary material are only displayed partially but also all figures are of very low quality, I was unable to review them and request a new version of the document.
Further points:
- The differences between “stromatolites”, “mats” and “stromatolite mats” is not clear to me. Are the “smooth mats” at Shark Bay actually “stromatolites” as suggested in the title?
- In the abstract the authors state that LARK001 has a “unique pattern in pigment distribution that likely reflect niche adaption”. I’m not sure what this means. Could the authors please explain what they consider as “pattern of distribution”?
- The structure of chlorophyll d should be described in more detail in line 37, e.g. by saying “in having a formyl group at C3 in ring A instead of a vinyl group” (which is also not a divinyl group). The following statement that the Qy band is shifted to 720 nm might be also misleading in this context and it might be better to mention that the Qy peak of the isolated chlorophylls is shifted from ~665 nm (chl a) to ~697 nm (chl d). Is there also a reference for the in vivo absorbance at 720 nm. According to Mohr et al. (2010, ISME J), the absorbance is around 707/710 nm. This also fits to the results displayed for LARK001 and MBIC11017 (Fig. 4B). (Correct also line 70 for the peak at 720 nm).
- The paragraphs starting in line 46 and line 56 seem to be duplicates, both describing the same strains.
- As far as I understand, A. marina was found in Antarctica but not in stromatolites as suggested in line 65. Since the description was not supported by 16S rRNA sequences this statement needs be also phrased more carefully.
- Why do the authors mention explicitly 440 nm for the blue region in line 69? The absorption spectra of chlorophyll d and a are different in this region.
- The importance of chlorophyll d in FaRLiP cyanobacteria should be further explained in line 76. Nürnberg et al. (2018, Science) and Gisriel et al. (2022, JBC) have shown that PSII contains one chlorophyll d in the reaction centre.
- For better reproducibility, the authors should provide more detail on the used materials and methods.
Line 118: Please specify how the mats were sectioned and what “thin” means.
Line 121: Specify what the spectral detection capabilities technically are.
Line 127: I find it difficult to get all the information from Fig. 1B. More details need to be added to this section to make the connection to the next parts clearer.
Line 130: What kind of “glass homogeniser”? Is there a reference to the media used in this study? How was the artificial seawater obtained and how much was added to BG11? What is the overall salt concentration of the media?
Line 131+139: Could the authors please explain the units of their light illumination? Why wasn’t “photon flux” in μmol m-2 s-1 chosen? I also find the difference in light intensity quite big. Could this explain the different ratio of chl d/a mentioned in the beginning?
Line 132: The temperature and the condition (shaking?) of incubation need to be specified.
Line 136: What was the concentration of agar and what type of agar was used?
Line 144: The microscopy technique which is mentioned in line 120ff seems to be the same as explained here.
Line 173: Further details on how the absorptions spectra were measured is needed.
Line 189: Was the sonication performed in a bath sonicator? Why were the samples vortexed three times? Was there a resting time phase in between to allow the samples to cool down again?
Line 198: What does the “1” stand for in front of method C?
Line 200: Could the authors please explain why the method was altered?
Line 202: The wavelengths could be ordered from short to long and it could be indicated for what they are used, e.g. 696 nm (chl d).
Line 217: Please explain “cyanobacterial broth” or rephrase.
- The description of the location in the results section (line 256ff) seems to be a duplication of the material and methods section and might be deleted or described differently.
- The confocal images of the mats shown in Figure 2 look great but unfortunately the quality is not good enough to see the spectra. The spectrum of the cells containing mainly chl a as mentioned in line 270 should be also added to Fig. 2A. What is the difference between Fig. 2A and B? Which parts are they from in the mat? Furthermore, the wavelengths of emission vary between the figure legend and the main text which makes it harder to locate them.
- It would be helpful if the authors could provide a “zoom in” view on the small fraction of organisms found in the late stage. Is it correct that different organisms were enriched in the late stage in comparison to the early stage? To avoid “contamination” by eukaryotic organisms the addition of cycloheximide is very helpful. It might be something the authors would like to include in later studies.
- The spectra shown for LarK001 in Fig. 4 A+B are interesting. Why do you see a shoulder at 735 nm? Since the lack of phycocyanin is a key difference between the strains, a spectrum with a better signal to noise ratio is required. This could also help to confirm that the 735 nm shoulder is a feature of this strain. Also, indicate which intensity axis belongs to the different strains in Figure 4E Why do you see here an additional fluorescence emission at 735 nm for MBIC11017 but not in Fig. 4C? This is also described in the figure legend in line 371: “peaks at 724 nm and 735 nm for MBIC11017”?
- Unfortunately, it is impossible to see the differences in pigment location in Figure 5D. Please provide a bigger image of the cells.
- The results on the aspect ratios and diameters should be further described in line 400.
- The results on the different growth rates are interesting. How where they measured? There is no detail on this in the material and methods section
- References should be added to the statement in line 419ff and the description of the chl d-containing photosystems in line 428ff.
- In the discussion further strains are mentioned based on cell morphology rather than the 16S rRNA results. Do the authors have evidence for a Chroococcidiopsis or Spirulina strain from their 16S sequences?
- References: Refs 2 and 3 are the same. Kuhl should be Kühl. Some journals are not correctly abbreviated, e.g. ref 21, 30,…
Finally, I would also like to strongly encourage the authors to deposit the strain in a culture collection to make it accessible for future studies and add the 16S rRNA sequence of LARK001 to the NCBI database.
Minor points (typos):
Line 21: What is “traditional” spectrometry? Just “spectrometry” would be okay.
Line 39: Change to “mats”.
Line 53: “characteristic (delete: a) small-subunit”
Line 53: check symbol. α?
Line 62: “recently” should be deleted
Line 75: change loci to locus
Line 102: delete “unique”
Line 144: no “:” after microscopy
Line 187: One hundred could be written here as “100”
Line 192: stored “at” -80°C
Line 195: Add “(Agilent)” after the name of the column
Line 204: Should be 16S rRNA analysis (no full stop needed; same for line 216)
Line 221f: Two sentences start with “Frist”. Please change
Line 288: of (delete: a) single colonies
Line 355: Change 7300 to 730
Line 364: add “LARK001” for the description of the strain
Line 370: differ (no s)
Line 372: Please avoid the use of the abbreviations “ex max” and “em max”
Line 404: under (delete: the) each light
Line 423: classic?
Line 429: conjugated with (delete: a) specialized… emission to (delete: the) 735 nm…
Line 508: What is considered as ancient here?
General: A. marina should be written in italics
Author Response
Dear Reviewer 1
We really appreciate your thoughtful comments and I believe we have addressed all of them. I believe the manuscript is in better shape as result of the recommended changes made the manuscript. Please find attached the point by point response as PDF
Kind regards
MIchael

Reviewer 2 Report
The manuscript describes the cultivation and thorough pigment analysis of a new strain of the Chl d-producing cyanobacterium Acaryochloris marina. The strain was isolated from a stromatolite at Shark Bay and is a valuable addition to this relatively poorly understood group of bacteria.
Re: Acaryochloris being still poorly described genetically and phenotypically (line 96), the authors can refer to a recently published paper that examines this issue for 37 A. marina strains (Microorganisms 2022, 10, 569; also part of this special issue). The authors could also cite this paper (and/or Ulrich et al. 2021 in Current Biology) to point out that MBIC11017 is still the only A. marina strain that produces phycocyanin (i.e., the type strain is the outlier, and the new strain is the norm).
For the phylogenetic analysis, please provide more details regarding the model of sequence evolution (i.e., substitution matrix, rate heterogeneity like gamma distribution or proportion of invariant sites) and how it was selected. For Fig. 1, why is strain MBIC11017 included twice in the phylogeny?
The authors should expand on the close relationship with strain HICR111A. For example, what is the % sequence identity? HICR111A has nitrogen fixation genes, so it is of interest whether the new strain can also fix nitrogen. Note that it has been shown (in Microorganisms 2022, 10, 569) that HICR111A likely belongs to a clade of A. marina that includes strains from the Caribbean, Japan and the South China Sea, all of which fix nitrogen.
I couldn’t find (or missed) the methods for growth rate measurement in the Methods, but growth rate was extraordinarily slow for the type strain (~10 days in white light) compared with in conventional liquid culture. So my guess is that this based on plate growth? Please clarify.
The Fv/Fm data in Fig 6B are completely redundant. I suggest removing the table.
Do you have any thoughts on the salinity tolerance of this strain?
Lines 474-490. This paragraph is not relevant to the manuscript.
The manuscript in its current state is a little sloppy. Several examples can be found in the minor comments below, but the authors need to thoroughly proofread the document.
Line 43 and throughout: check spacing.
Line 48 and throughout. Check use of italics.
Line 54. Should be “b-proteobacterium”. The statement that “Acaryochloris is a proteobacterial-cyanobacterial hybrid entity” may be misconstrued by readers. It should be pointed out specifically that it is the 16S rRNA molecule that is being referred to here.
Lines 59-61. This is an example of redundant text. The details of isolation of Awaji and CCMEE 5410 strains were already discussed earlier in the Introduction.
Line 75: “Locus”.
Line 182: “Raw data were exported…”
Line 204: “SSU rRNA”.
Line 253-255. Remove boilerplate language from the publisher. (Also for the Author Contributions section).
Line 299: “chl”.
Line 315 and throughout: Check missing prepositions. “presence Halomicronema”.
Line 423: “Species”.
Line 450: “Pacific… ecological niche”
Throughout: consistency on italics for Chl d, A. marina etc.
Fig. 2B. The arrow appears white to me rather than yellow.
Author Response
Dear Reviewer 2
We really appreciate your thoughtful comments and I believe we have addressed all of them. I believe the manuscript is in better shape as result of the recommended changes made the manuscript. Please find attached the point by point response as PDF
Kind regards
MIchael

Round 2
Reviewer 1 Report
Dear authors,
many thanks for addressing the comments so thoroughly.
I just like to point out that in the "Author contributions" sections the guidelines for the authors are still mentioned. They should be removed from lines 557-558 and 568-570.
Furthermore, a zoom in view of the cells shown in Figure 5D should be provided to see the differences in pigment loocation, as requested earlier.
Author Response
Dear reviewer 1, thanks for pointing out the "guidelines" still being evident in the author contributions. I have now removed this and I have also provided the zoom of the confocal images as requested
Kind regards
Michael
